# Effect of post-implant exercise on tumour growth rate, perfusion and hypoxia in mice

Linda A. Buss[1], Abel D. Ang[1], Barry Hock[2], Bridget A. Robinson[1,3], Margaret J. Currie[1], Gabi U. Dachs[1]*

1 Mackenzie Cancer Research Group, Department of Pathology and Biomedical Science, University of Otago, Christchurch, New Zealand, 2 Hematology Research Group, Department of Pathology and Biomedical Science, University of Otago, Christchurch, New Zealand, 3 Canterbury Regional Cancer and Hematology Service, Canterbury District Health Board, Christchurch, New Zealand

* gabi.dachs@otago.ac.nz

**Data Availability Statement:** All relevant data are within the manuscript and its Supporting Information files.

**Funding:** This research was funded in part by the Mackenzie Charitable Foundation, NZ (part salaries

## Abstract

Preclinical studies have shown a larger inhibition of tumour growth when exercise begins prior to tumour implant (preventative setting) than when training begins after tumour implant (therapeutic setting). However, post-implantation exercise may alter the tumour microenvironment to make it more vulnerable to treatment by increasing tumour perfusion while reducing hypoxia. This has been shown most convincingly in breast and prostate cancer models to date and it is unclear whether other tumour types respond in a similar way. We aimed to determine whether tumour perfusion and hypoxia are altered with exercise in a melanoma model, and compared this with a breast cancer model. We hypothesised that post-implantation exercise would reduce tumour hypoxia and increase perfusion in these two models. Female, 6–10 week old C57BL/6 mice were inoculated with EO771 breast or B16-F10 melanoma tumour cells before randomisation to either exercise or non-exercising control. Exercising mice received a running wheel with a revolution counter. Mice were euthanised when tumours reached maximum ethical size and the tumours assessed for perfusion, hypoxia, blood vessel density and proliferation. We saw an increase in heart to body weight ratio in exercising compared with non-exercising mice (p = 0.0008), indicating that physiological changes occurred with this form of physical activity. However, exercise did not affect vascularity, perfusion, hypoxia or tumour growth rate in either tumour type. In addition, EO771 tumours had a more aggressive phenotype than B16-F10 tumours, as inferred from a higher rate of proliferation (p<0.0001), a higher level of tumour hypoxia (p = 0.0063) and a higher number of CD31+ vessels (p = 0.0005). Our results show that although a physiological training effect was seen with exercise, it did not affect tumour hypoxia, perfusion or growth rate. We suggest that exercise monotherapy is minimally effective and that future preclinical work should focus on the combination of exercise with standard cancer therapies.

to ADA, MJC, GUD). The funder had no role in study design, data collection and analysis, decision to publish, or preparation of the manuscript.

**Competing interests:** The authors have declared that no competing interests exist.

## Introduction

It is now well-established that exercise or physical activity is efficacious in a number of different ways at different time-points along the cancer continuum. Pre-diagnosis physical activity reduces the risk of developing a range of different cancers, including breast cancer [1]. One apparent exception is that the risk of developing malignant melanoma is increased with leisure-time physical activity, but this loses significance when adjusted for UV radiation exposure [1]. A large body of data indicates that during treatment, exercise can help to reduce treatment-related side-effects, improve symptoms of anxiety and depression, and improve health-related quality of life (reviewed in [2]). In terms of survival outcomes, post-diagnosis exercise reduces breast cancer mortality by approximately 40% (systematically reviewed in [3]). On the other hand, clinical data on melanoma are sparse, there being (to our knowledge) only one study investigating the effect of exercise on survival in melanoma patients. In that study, the authors found that pre-diagnosis exercise did not affect survival outcomes in patients with high-risk primary melanoma [4]. However, survival rates for primary melanoma are very high regardless (>90%, [5]), which means there is little room for improvement. It is unknown whether post-diagnosis exercise affects survival outcomes in either primary or metastatic melanoma patients.

In preclinical studies, varying and conflicting data have been reported. Although there have been numerous reports of the effects of exercise on tumour progression in rodents [reviewed in 2,6–8], only one systematic review includes a subgroup analysis to compare the effects of pre- vs post-implantation exercise [8]. This is an important distinction, as pre-implantation exercise preconditions the animal and mimics the use of exercise as a preventative measure, while post-implantation exercise mimics the use of exercise in a therapeutic setting (post-diagnosis). When studies were grouped according to the timing of exercise initiation (pre-implant, post-implant or both pre- and post-implant), there was a significant difference between groups (p = 0.0284), with exercise starting before implant and continuing throughout moderately reducing final tumour size and exercise post-implant having only a small anti-tumour effect [8]. There was only one study investigating exercise pre-implant; which found a non-significant increase in tumour size with exercise [8]. This suggests that the biggest impact is seen when exercise begins prior to tumour implant and continues post-implant. However, it is important to determine whether post-implantation exercise can also be effective, in order to inform clinicians faced with sedentary patients.

Tumour vascularity is an integral determinant of how the tumour develops and progresses, and an adequate blood flow providing nutrients and oxygen is required. Therefore, when cells become hypoxic, a variety of pro-survival genes are turned on, including genes responsible for stimulating angiogenesis [9]. However, this does not result in well-organised and mature vascular networks as it would in normal tissue, but rather in chaotic, dysfunctional vessel systems [10]. This perpetuates tumour hypoxia, which contributes to an aggressive tumour phenotype (increased metastatic potential and treatment resistance) and is a negative prognostic indicator [11]. Many studies have attempted to reduce or exploit tumour hypoxia, including hyperbaric oxygen chambers, hypoxia-activated pro-drugs and hypoxic cell radiosensitisers, but with limited success [12]. Nevertheless, addressing tumour hypoxia remains an attractive strategy due to the potential for large beneficial effects.

Vascular normalisation is another approach to reducing hypoxia and improving drug delivery to the tumour. Here, the goal is to encourage growth of more functional and homogenously distributed tumour vessels, resulting in more evenly perfused tumour tissue. A few rodent studies in orthotopic breast and prostate tumours, as well as subcutaneous pancreatic

tumours, have found that exercise can increase tumour perfusion [13–15], vascularity [15,16] and/or reduce hypoxia [15,17], thus 'normalising' the tumour microenvironment.

In preclinical melanoma, post-diagnosis exercise did not change CD31[+] vessel density in B16-F10 tumours, although doxorubicin delivery to the tumour was enhanced, suggesting improved blood flow [14]. The anti-cancer effect of exercise in melanoma has largely been attributed to improved immune function, with pre-implantation exercise increasing T cells, natural killer (NK) cells and dendritic cell numbers [18]. To our knowledge, no study has investigated how exercise affects hypoxia in melanoma.

Our preclinical study aimed to investigate the effect of short-term, post-implantation exercise in preclinical models of breast cancer and melanoma. We hypothesised that, with post-implantation exercise, tumours would exhibit increased tumour perfusion and reduced hypoxia, albeit with limited effect on tumour growth rate. To investigate this, we inoculated female C57BL/6 mice subcutaneously with B16-F10 or orthotopically with EO771 tumour cells and provided them with a running wheel. Immunohistochemical techniques were used to evaluate hypoxia, perfusion and vascularity, as well as tumour cell proliferation.

## Materials and methods

### Materials

EO771 breast cancer cells were kindly gifted by Dr Andreas Moeller (QIMR Berghofer, Australia) and B16-F10 melanoma cells were sourced from American Type Culture Collection, Cryosite Distribution, Australia. Pimonidazole was from Hypoxyprobe™-1 (Burlington, Massachusetts, USA). General chemicals, bovine serum albumin (BSA), phosphate buffered saline (PBS) and Hoechst 33342 were from Sigma Aldrich (St Louis, MO, USA). Antibodies against mouse cluster of differentiation 31 (CD31) phosphohistone H3 (pHH3) and pimonidazole are shown in Table 1. The REAL EnVision Detection System, Peroxidase/DAB+, Rabbit/ Mouse (K5007, Dako, Copenhagen, Denmark) was used for immunohistochemical staining.

### Mouse model and exercise setup

Ethical approval for this study was obtained from the University of Otago Animal Ethics Committee (C04/17 and C01/16) and international guidelines on animal welfare were followed [19]. Female C57BL/6 mice were bred in-house and maintained on a 12:12 hour light-dark cycle. Mice were fed a standard chow diet; food and water was provided *ad libitum*.

Mice were housed in pairs in rat cages (floor area 904 cm$^2$) with a perforated cage divider to allow mice to see, smell and hear each other while preventing physical contact. This arrangement was designed to minimise isolation stress whilst allowing measurement of individual mouse running distance. Modified exercise wheels (Fast-Trac Saucer Wheel, Bio-Serv,

**Table 1. List of antibodies.**

| Antigen | Clonality | Host species | Supplier | Antibody registry ID | Catalogue number | Clone number | Conjugation | Dilution |
|---------|-----------|--------------|----------|----------------------|------------------|--------------|-------------|----------|
| CD31 | Polyclonal | Rabbit | Abcam, Cambridge, UK | AB_2802125 | ab124432 | NA | None | 1:5000 |
| pHH3 | Polyclonal | Rabbit | Abcam | AB_304763 | ab5176 | NA | None | 1:10 000 |
| Pimonodazole | Monoclonal | Mouse | Hypoxyprobe Inc., Massachusetts, USA | AB_2801307 | HP FITC MAb-1 | 4.3.11.3 | FITC | 1:500 |

CD31: Cluster of differentiation 31; pHH3: phosphohistone H3; FITC: fluorescein isothiocyanate.

Flemington, NJ, USA) fitted with magnetic sensors to count revolutions were provided to quantify running distance (Decision Consulting Ltd, Christchurch, NZ) [20].

## Tumour model

B16-F10 melanoma were used up to passage 6 and EO771 breast cancer cells were used up to passage 16. Mycoplasma testing is routinely performed in our laboratory.

At 6–10 weeks of age, mice were injected with either $1x10^6$ B16-F10 cells in 50 μL sterile PBS subcutaneously into the shaved right flank, or $2x10^5$ EO771 cells in 20 μL sterile PBS into the $4^{th}$ mammary fat pad. Mice were divided into two groups: exercise (Ex, n = 12 for each tumour type), receiving an exercise wheel on the day of tumour implant, or no exercise (No Ex, n = 12 for each tumour type). Wheels were provided on the day of implant to allow a few days for mice to acclimatise to the wheel before tumours became detectable, in order to maximise exercise effects during the short post-implant timeframe.

Tumours were measured daily by calliper and tumour volume was estimated using the following formula: tumour volume = $width^2$ x (length/2). When tumours reached a maximum of $1000 \text{ mm}^3$ (B16-F10) or $600 \text{ mm}^3$ (EO771), mice were injected intraperitoneally with 60 mg/kg pimonidazole 90 minutes before euthanasia (to visualise tumour hypoxia [21]) and with 60 μL of 5 mg/mL Hoechst 33342 intravenously 1 minute before euthanasia (to visualise perfused tumour vessels [15]). Mice were euthanised by isoflurane overdose and cervical dislocation. Mice were euthanised early if the presence of intra-peritoneal tumours was suspected (breast cancer only, n = 3) or ulceration of the tumour occurred (B16-F10 n = 4, EO771 n = 2). These mice were not included in the survival analysis. Mice with ulcerated tumours were included in the histological analyses, but those with intra-peritoneal tumours were not. Additionally, one exercising mouse bearing an EO771 tumour was euthanised early and excluded from analysis due to pyometra. Tumours, livers, kidneys, hearts and spleens were removed, weighed and cut into fragments. One third of the tumour was frozen at -80˚C, one third was OCT embedded and frozen at -80˚C along with half the spleen and part of the liver, and one third was formalin-fixed and paraffin embedded (FFPE) along with the other half of the spleen and part of the liver. All analyses on harvested tissue samples were blinded to treatment.

## Immunohistochemistry

FFPE blocks were cut into 3 μm sections. Before staining, slides were baked at 60˚C for at least one hour, then deparaffinised with xylene and rehydrated through graded ethanol baths and MilliQ water. Antigen retrieval was performed by boiling sections in a pressure cooker for 3 minutes at full pressure using citrate buffer (10 mM tri-sodium citrate, 0.05% Tween-20 (v/v), pH 6.0). Sections were stained using the REAL EnVision Detection System, and probed with antibodies against either CD31 or pHH3 overnight at 4˚C.

pHH3 staining was quantified by calculating the percentage of pHH3 positive cells per visual field at 20x magnification, and averaging over 10 random fields. CD31 staining was quantified by counting the number of $CD31^+$ vessels per 20x field and averaging over 10 random fields. In a number of melanomas, dark pigmentation was observed, making true staining difficult to identify and quantification unreliable. For this reason, six tumours were excluded from the CD31 analysis.

## Immunofluorescence

OCT-embedded frozen samples were cut into 8 μm sections and stored at -20˚C. Before staining, sections were dried at room temperature for 30 minutes before fixing in 10% neutral-

buffered formalin for 10 minutes. Sections were blocked with 10% BSA and probed with a FITC-conjugated pimonidazole antibody overnight at 4°C.

Image J software was used to determine the hypoxic area at 5x magnification, using 1–5 images as necessary to cover the entire section. Hypoxic area was then expressed as a percentage of total tumour area (necrotic areas were excluded from the analysis).

Hoechst 33342 staining was quantified by counting the number of Hoechst 33342+ vessels per 10x field and averaging over 10 random fields.

### Statistical analysis

All data were analysed using GraphPad Prism 7. The D'Agostino and Pearson normality test was used to determine if data followed a normal distribution. Accordingly, differences between groups were tested using the two-tailed student's t test for normally distributed data, and the two-tailed Mann-Whitney rank test for non-parametric data, as indicated in the figure legends. Correlations were tested for using either Pearson (for normally distributed data) or Spearman (for non-parametric data) correlation. Differences in variance were tested for using the F test. Survival analysis was performed using the Log-rank (Mantel-Cox) text. P values less than 0.05 were considered significant.

## Results

### Tumour growth rate

Exercising mice (Ex) received a running wheel, while non-exercising controls (No Ex) did not. Tumour volume was estimated daily using calliper measurement. The median time to endpoint was 17 days for mice bearing melanomas and 21 days for mice bearing breast tumours. We saw no difference in tumour growth rate between mice with or without access to a running wheel for both B16-F10 and EO771 tumours (Fig 1C and 1D, S1 Fig), and there was no difference in survival (Fig 1A and 1B). In addition, average daily running distance was not correlated with tumour growth rate (S2A and S2B Fig). One exercising mouse with B16-F10 melanoma survived much longer than all the others, but this did not appear to be due to any of the measured microenvironmental factors or mouse characteristics.

### Cohort characterisation

Mice with B16-F10 melanoma and mice with EO771 breast cancer ran an average of 8 km per day across the duration of the study, although this was subject to large inter and intra-individual variation (range: <1 km/day– 23 km/day; Fig 2). Average daily running distance was steady throughout, indicating that tumour burden did not affect activity towards the end of the study and that the systemic effects are comparatively low. There also was no major difference between mice with tumours grown subcutaneously on the back (B16-F10) and tumours grown orthotopically in the mammary fat pad (EO771), similarly demonstrating that tumour burden and location was not affecting ability to exercise using a wheel. The distance run by mice in this study is similar to previously published reports of running distance in young, female C57BL/6 mice without tumours [22].

Most mice lost weight in the first few days after tumour implant (S3A–S3D Fig). Weight returned to baseline or increased by the end of the study in 50% of mice, but remained below starting weight in the other 50% of mice (Table 2), regardless of tumour type or exercise group. Using a cut-off of 5% body weight loss, five mice with B16-F10 melanoma can be described as having developed cachexia (No Ex: n = 2, Ex: n = 3) and two mice with EO771 breast cancer (No Ex: n = 1, Ex: n = 1). A cut-off of 10% weight loss has also been described for

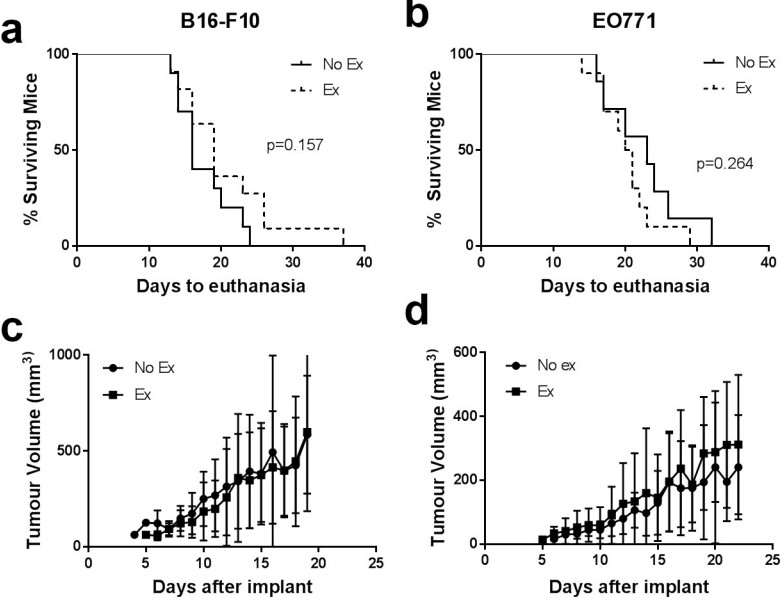

**Fig 1. Tumour growth rate and mouse survival are unaffected by exercise in B16-F10 and EO771 tumours.**
Survival curves for mice bearing B16-F10 (a) or EO771 (b) tumours (endpoint due to tumour size only, mice euthanised due to other endpoints were excluded from the survival analysis). Average tumour growth rate in mice bearing B16-F10 (c) or EO771 tumours (d). Exercising mice (Ex) received a running wheel, while non-exercising controls (No Ex) did not. Data are shown as mean ± SD. n = 3–12 for B16-F10 or n = 3–10 for EO771 (progressively fewer mice over time as they were euthanised).

cancer cachexia in mice [23], but given that other symptoms of cachexia (such as anorexia) begin prior to a noticeable weight loss in mice [24], we chose to use a more conservative cut-off.

The spleen, liver, both kidneys and heart were removed and weighed after euthanasia. Organ weights were normalised to final body weight (minus tumour weight) for analysis. Hearts from exercising mice with EO771 tumours were significantly heavier than those from non-exercising mice (p = 0.0008, Table 2) and a similar trend was seen for mice with B16-F10 tumours (p = 0.095, Table 2). No other organ weights were significantly different between exercising or non-exercising mice, for either tumour type (Table 2).

In mice that had lost weight over the course of the study (including cachectic mice), EO771 tumours grew more rapidly to palpable size (100 mm³, defined as lag phase of tumour growth),

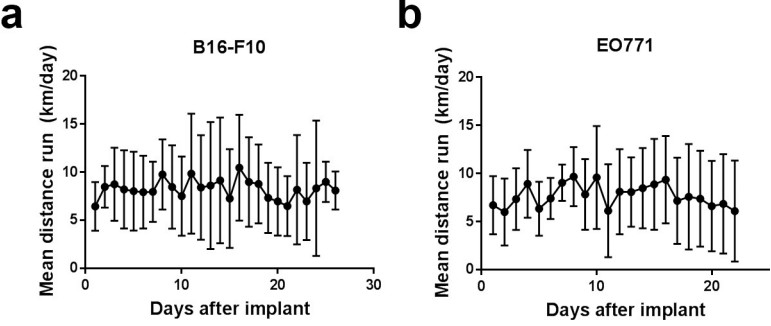

**Fig 2. Running distance of tumour-bearing mice.** Average daily running distance on a running wheel by mice bearing B16-F10 **(a)** or EO771 **(b)** tumours. Data are shown as mean ± SD; n = 3–12 (progressively fewer mice over time as they were euthanised).

**Table 2. Body and organ weights of non-exercising vs exercising mice with B16-F10 or EO771 tumours.**

|  | B16 No Ex | B16 Ex | p-value | EO771 No Ex | EO771 Ex | p-value |
|---|---|---|---|---|---|---|
| Initial body weight (g) | 20.0±1.85 | 19.2±1.68 | 0.293 | 18.8±1.27 | 18.3±1.38 | 0.344 |
| Final body weight (g) | 20.3±1.04 | 19.3±1.56 | 0.0615 | 18.8±0.89 | 18.4±1.10 | 0.271 |
| Change in body weight (%) | 1.45±5.22 | 1.39±6.66 | 0.980 | 0.32±3.94 | 0.68±4.07 | 0.832 |
| Heart/body weight (mg/g) | 5.95±0.46 | 6.45±0.88 | 0.095 | 6.13±0.52 | 6.98±0.52 | 0.0008*** |
| Spleen/body weight (mg/g) | 6.98±6.27 | 4.73±0.88 | 0.232 | 5.22±1.28 | 4.67±0.78 | 0.260 |
| Liver/body weight (mg/g) | 45.1±5.96 | 46.8±6.07 | 0.494 | 46.8±6.08 | 50.4±3.11 | 0.099 |
| Kidney/body weight (mg/g) | 14.5±1.40 | 14.7±1.05 | 0.615 | 15.7±0.74 | 15.9±1.02 | 0.654 |

Values are means±SD. p-values are for exercising (Ex) vs non-exercising (No Ex) control for the respective tumour types. Data were analysed using a two-tailed student's t test; n = 10–12.

than those whose weight remained stable or who gained weight (S4B Fig), regardless of whether mice exercised or not. There was no difference in lag phase growth (time to reach 175 mm$^3$) in mice with B16-F10 tumours (S4A Fig). Exponential tumour growth rate (time for the tumour to quadruple in volume) was unchanged by weight loss in both tumour types (S4C and S4D Fig). It is noteworthy that mice that lost weight while bearing EO771 tumours had poorer overall survival (time to euthanasia due to tumour burden) than those that did not lose weight (p = 0.0008, S4F Fig), while those bearing B16-F10 tumours had similar survival regardless of weight change (S4E Fig). It is unclear whether accelerated tumour growth is causing the weight loss or whether weight loss supports more rapid tumour growth.

## Tumour hypoxia, perfusion, vascularity and proliferation

Previous reports in murine breast tumours have indicated that post-implantation exercise can increase tumour perfusion and vascularity, and reduce hypoxia, compared with non-exercising mice [15]. We aimed to confirm this in EO771 breast tumours and determine whether it holds true for melanoma. We observed that perfused vessels (according to Hoechst 33342 staining) segregated well from hypoxic areas (stained for pimonidazole adducts), with very little overlap, confirming adequate oxygen delivery through perfused vessels (Fig 3A). In addition, we confirmed that Hoechst 33342 staining co-localises with CD31$^+$ blood vessels, but that not all CD31$^+$ vessels are perfused (S5 Fig). We found that hypoxic area and perfused vessel number were unchanged in tumours from exercising compared with non-exercising mice, for both B16-F10 and EO771 tumours (Fig 3A–3E). Hypoxia was more variable between tumours than within individual tumours (mean SD of hypoxic fraction of individual fields used for analysis: 4.96 and 7.8 for B16-F10 and EO771, respectively, vs SD of mean hypoxic fraction between tumours: 7.27 and 9.69 for B16-F10 and EO771, respectively). However, it was noteworthy that there was significantly less variation between EO771 tumours in the number of perfused vessels from exercising compared with non-exercising mice (F test, p = 0.024, Fig 3E). A similar trend was seen in B16-F10 tumours, but the effect was much less pronounced (Fig 3C).

There was no difference in CD31$^+$ vessel density in tumours from exercising compared with non-exercising mice, for both B16-F10 and EO771 tumours (Fig 4). Similarly, average daily running distance was not correlated with either the number of perfused vessels, tumour hypoxia or the number of CD31$^+$ vessels (S2C–S2H Fig).

The area of tumour hypoxia did not correlate with either the total number of vessels or the number of perfused vessels in either tumour type (S6 Fig). Likewise, the total number of vessels did not correlate with the number of perfused vessels in either tumour type (S6 Fig).

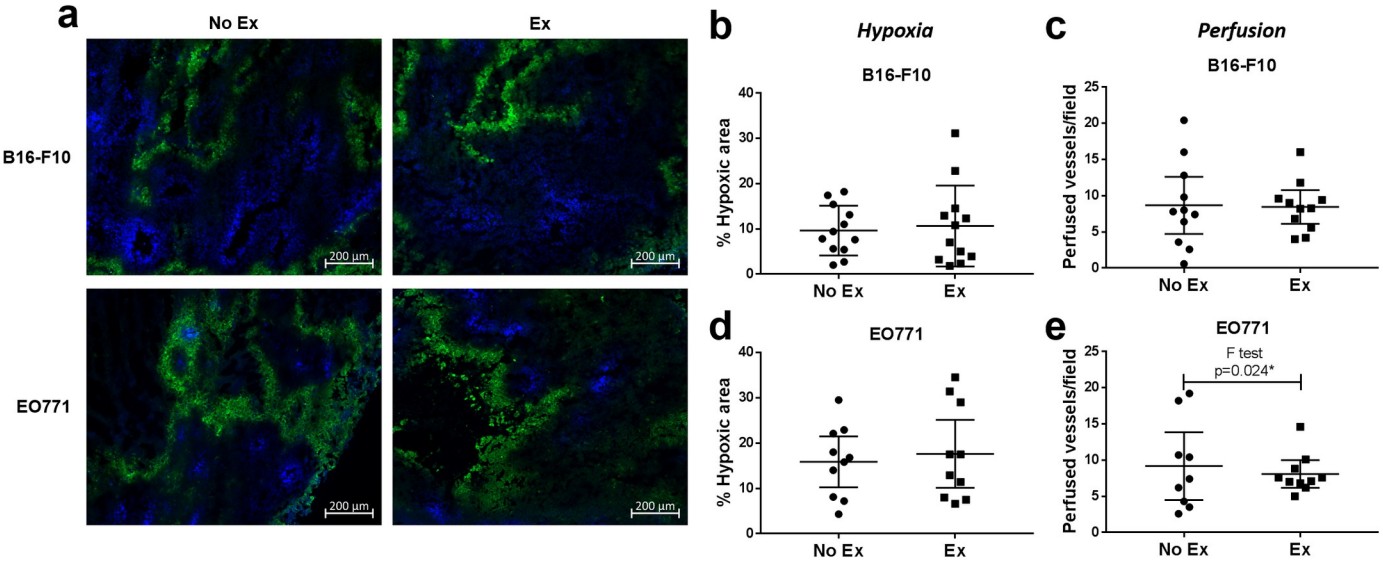

**Fig 3. Exercise reduces variance in perfusion in EO771 tumours. (a)** Representative immunofluorescent images of sections containing Hoechst 33342 (perfused blood vessels, blue) and stained for pimonidazole (hypoxia, green) in B16-F10 and EO771 tumours from non-exercising vs exercising mice. Quantification of hypoxic area in B16-F10 **(b)** and EO771 **(d)** tumours from non-exercising vs exercising mice. Quantification of perfused blood vessels in B16-F10 **(c)** and EO771 **(e)** tumours from non-exercising vs exercising mice. B16-F10 No Ex and Ex: n = 12, EO771 No Ex: n = 9 (two mice had intra-peritoneal tumours and one tumour exhibited low-level, diffuse perfusion which could not be quantified) and Ex: n = 10 (one mouse had intra-peritoneal tumours, and one was euthanised before tumour development due to pyometra). Difference in variance analysed using the F test. Data are presented as individual data points and mean ± 95% CI.

Tumour cell proliferation was measured by immunohistochemical staining for pHH3, a mitotic marker, and by calculating the percentage of pHH3 positive nuclei. No difference was

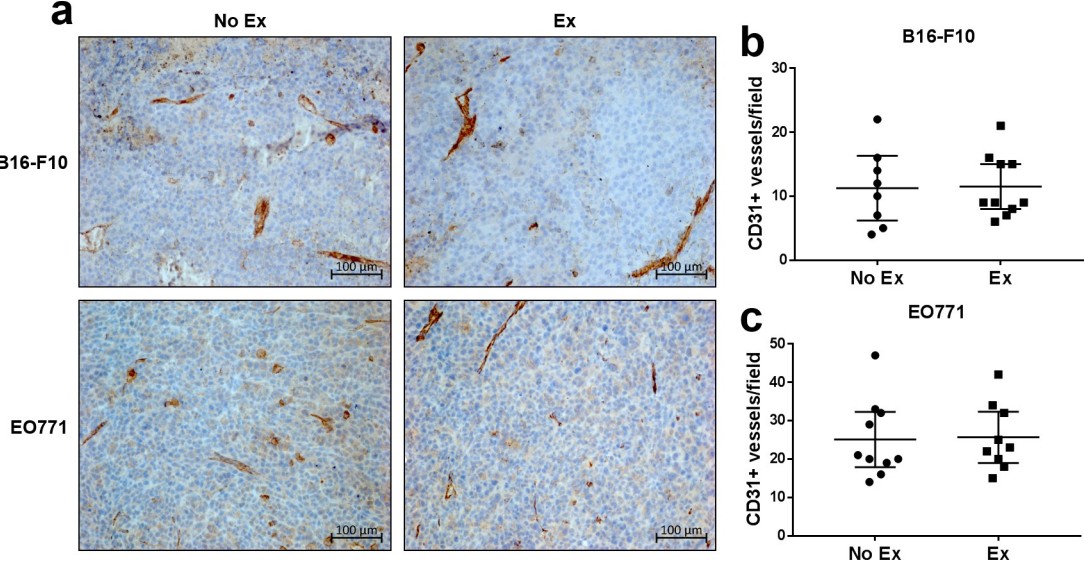

**Fig 4. Exercise does not change CD31$^+$ vessel density in B16-F10 or EO771 tumours. (a)** Representative immunohistochemical images of B16-F10 and EO771 tumours from exercising vs non-exercising mice. Quantification of the number of CD31$^+$ vessels in B16-F10 **(b)** and EO771 **(c)** tumours from non-exercising vs exercising mice. B16-F10 No Ex: n = 8 (four tumours could not be quantified due to dark pigmentation), B16-F10 Ex: n = 10 (two tumours could not be quantified due to dark pigmentation), EO771 No Ex: n = 10 (two mice had intra-peritoneal tumours) and Ex: n = 10 (one mouse had intra-peritoneal tumours, and one pyometra). Data are presented as individual data points and mean ± 95% CI.

observed in the proliferation of tumours from exercising versus non-exercising mice for both B16-F10 and EO771 tumours (Fig 5A–5C).

Taken together, our data show that exercise beginning after tumour implantation does not alter the mean level of tumour hypoxia, perfusion, CD31$^+$ vessel density or cancer cell proliferation. However, inter-tumour perfusion heterogeneity was reduced with exercise in EO771 tumours.

## Comparison of B16-F10 and EO771 tumours

We compared microenvironmental features between B16-F10 and EO771 tumours to determine differences between the two tumour models. As no differences were seen with exercise in any of the investigated features, we pooled results from non-exercising and exercising mice for each tumour type.

EO771 tumours were significantly more proliferative than B16-F10 tumours (p<0.0001, Fig 6A). EO771 tumours were also significantly more hypoxic (p = 0.0063) and had a higher CD31$^+$ vessel density (p = 0.0005) than B16-F10 tumours (Fig 6B and 6C), while the number of perfused vessels per field were similar in EO771 compared with B16-F10 tumours (Fig 6D).

These results suggest that EO771 tumours have a more aggressive phenotype than B16-F10 tumours, characterised by higher levels of tumour hypoxia, more CD31$^+$ vessels (but no increase in perfusion) and more proliferative tumour cells.

## Discussion

We found that short-term, post-implantation exercise did not alter tumour growth rate, hypoxia, perfusion, blood vessel density or cell proliferation in either tumour type in a murine model. Mice bearing EO771 tumours who lost weight had a shorter tumour lag growth phase and poorer survival than those who did not lose weight. Additionally, we observed that EO771 tumours had a more aggressive phenotype than B16-F10 tumours, with increased levels of tumour cell proliferation, hypoxia and CD31$^+$ vessel density. This study is the first to investigate changes in tumour hypoxia and perfusion with post-implantation exercise in melanoma, and to compare these aspects of the tumour microenvironment between two different tumour models in exercising mice.

There is much discussion as to which mouse exercise modality is best: forced modalities such as treadmill running and swimming allow better control of exercise dosage, but are inherently stressful [25,26]. A recent review article argues that because mice naturally run far more than most humans are physically capable of, voluntary wheel running experiments cannot lead to human relevant data [38]. We argue that exercise doses that elicit a physiological response will naturally be different between the two species, but this does not mean that mouse exercise studies cannot provide useful mechanistic data in the exercise oncology setting. Indeed, we observed a significant increase in heart to body weight ratio in exercising mice with EO771 tumours, and a similar trend in mice with B16-F10 tumours. Exercise-induced cardiac hypertrophy is a well-established phenomenon in humans [27]. In addition, healthy female mice exposed to 21 days of voluntary wheel running had a significantly higher heart to body weight ratio than their non-exercising counterparts [28]. This also holds true in tumour-bearing mice. Sturgeon *et al.* found that mice bearing B16-F10 melanoma and exposed to 16 days of treadmill running had significantly higher heart to body weight ratios than their non-exercising counterparts [29]. Together, this indicates that in mice, heart weight increases within a week or two of exercise, and this remains true in tumour-bearing mice. Thus, our cohort showed the expected increase in cardiac size, demonstrating that our exercise protocol was having a physiological effect.

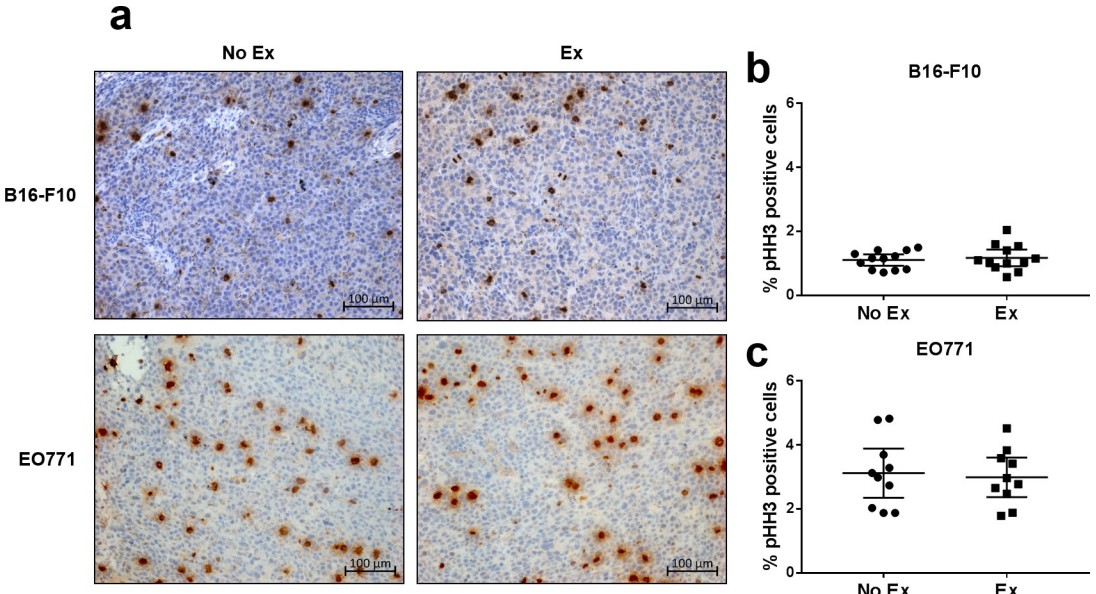

**Fig 5. Exercise does not change tumour cell proliferation in B16-F10 or EO771 tumours. (a)** Representative immunohistochemical staining for pHH3, a mitotic marker, in tumour sections from exercising vs non-exercising mice with B16-F10 melanoma or EO771 breast cancer. Quantification of the percentage of pHH3$^+$ cells in B16-F10 **(b)** and EO771 **(c)** tumours from exercising vs non-exercising mice. B16-F10: n = 12 per group. EO771: n = 10 per group (three mice had intra-peritoneal tumours, and one pyometra). Data are shown as individual data points and mean ± 95% CI. Data analysed using a two-tailed student's t test.

We observed no change in tumour growth rate in exercising vs non-exercising mice, regardless of tumour type. For B16-F10 melanoma, this is in agreement with Pedersen *et al.*, who found that wheel running beginning at tumour implant did not alter tumour growth rate, although exercise beginning 4 weeks prior to tumour implant significantly slowed tumour growth [18]. In two previous studies in mice exercise beginning at tumour implant significantly slowed orthotopic EO771 tumour growth, but this difference was small (tumour volume at endpoint approx. 1300–1500 mm$^3$ for non-exercising mice and 900 mm$^3$ for exercising mice) and only became apparent once tumours exceeded 600 mm$^3$ (i.e. there was no difference in growth rate up to a tumour size of 600 mm$^3$ [15,30]), which was the maximum ethical size used in our study. Thus, our study is consistent with previous data for EO771 tumours up to 600 mm$^3$.

Most preclinical studies that reported a statistical reduction in tumour growth rate with post-implantation exercise show only a marginal slowing of tumour growth [15,31–33]. A recent systematic review and meta-analysis found a "small to moderate" effect size for exercise to reduce final tumour size [8]. However, of the 8 studies included that showed a statistically significant difference in tumour size, one of these showed an increase with exercise, one had a small effect size and four had a 'probably high' risk of bias [8]. As there is little consistency between studies in terms of the effect of post-implantation exercise on tumour growth, it seems unlikely that exercise as a sole intervention (monotherapy) has a meaningful effect.

Previous studies in orthotopic breast (4T1 and EO771) and prostate tumours have reported a reduction in tumour hypoxia [15,17], increase in tumour perfusion [13–15] and increase in CD31$^+$ vessel density with post-implantation exercise [15]. In contrast, we found no change in the mean value of any of these parameters with exercise, in either B16-F10 or EO771 tumours. For B16-F10 melanoma, this could be due to differences in tumour type and location

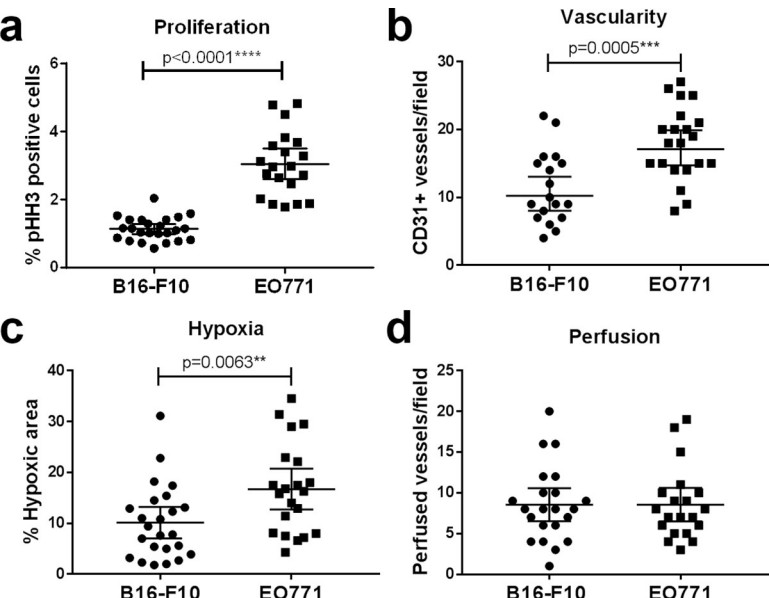

**Fig 6. EO771 tumours are more proliferative, more vascular and more hypoxic than B16-F10 tumours.** Exercising and non-exercising mice were pooled for this analysis as no difference was seen with exercise. **(a)** Quantification of the percentage of pHH3+ cells in B16-F10 vs EO771 tumours. B16-F10: n = 24, EO771: n = 20 (three mice had intra-peritoneal tumours, and one pyometra). **(b)** Comparison of the number of CD31+ vessels in B16-F10 and EO771 tumours. B16-F10: n = 18 (six tumours could not be quantified due to dark pigmentation), EO771: n = 20 (three mice had intra-peritoneal tumours, and one pyometra). Quantification of hypoxia **(c)** and perfusion **(d)** in B16-F10 vs EO771 tumours. B16-F10: n = 24, EO771: n = 20 (three mice had intra-peritoneal tumours, and one pyometra). Data analysed using two-tailed Mann-Whitney test. (Exercising and non-exercising animals were pooled for this analysis). Data analysed using two-tailed student's t test. Data are presented as individual data points and mean ± 95% CI.

(subcutaneous melanoma vs orthotopic breast and prostate cancer). Garcia *et al.* found that, during exercise, blood flow was increased to orthotopic prostate tumours in rats, but decreased to subcutaneous prostate tumours using the same cell line and rat strain [34]. This was paralleled by decreased blood flow to subcutaneous adipose tissue and skin (i.e. the tissues adjacent and attached to the subcutaneous tumour). Therefore, although tumour vessels themselves are less able to respond to haemodynamic cues than normal vessels [29], the response of the surrounding tissue to exercise seems to play an important role in regulating blood flow.

In the case of EO771 breast tumours, those grown in our study were smaller than those in the earlier study by Betof *et al.* [15], which may explain the lower level of hypoxia observed in our study and may influence the physiological response of the tumour to exercise.

There was significantly less variation in perfusion between EO771 tumours from exercising compared with non-exercising mice, and this trend was also seen in B16-F10 tumours. This suggests that exercise may improve the regulation of vascular maturation, particularly in EO771 tumours. This supports previous work which reported more homogenous perfusion across orthotopic breast [15] and prostate [13] tumours with exercise.

In normal tissue, blood flow is locally regulated by contraction and dilation of arterioles. However, tumour vessels exhibit not only lower contractility upon noradrenergic stimulation compared with normal vessels [34,35], but also reduced responsiveness to vasodilators [36]. Together, this reflects the impaired ability of tumour vessels to respond to haemodynamic cues. In addition, oxygen delivery is determined not only by blood flow, but also by additional haemodynamic properties such as capillary transit time and mean transit time, which are higher in tumour vasculature, likely reducing oxygen extraction [37,38]. This may explain the

lack of association between tumour perfusion and hypoxia seen by us and others [17], and the differing effects of exercise on tumour perfusion across different studies. More research is required to gain a full picture of tumour haemodynamics relating to tissue perfusion both during acute exercise and following a training period (chronic exercise).

The main limitation of our study is that both B16-F10 and EO771 tumours grow very rapidly, limiting the length of time available for exercise to effect changes on the tumour microenvironment. Furthermore, we were unable to measure dynamic changes in perfusion and hypoxia in the whole tumour.

We conclude that exercise as a monotherapy post-implant may have very limited effects on tumour growth. A small number of studies have used exercise in combination with conventional cancer therapies and demonstrated potentiation of the effect of the accompanying therapy, even in the absence of an exercise-only effect on tumour growth [14,15,29,39]. As this also reflects a more clinically relevant scenario, future preclinical studies should focus on the combination of exercise with other treatments such as chemotherapy, radiotherapy or immunotherapy.

## Supporting information

**S1 Fig. Individual tumour growth curves for non-exercising and exercising mice bearing B16-F10 or EO771 tumours.** B16-F10: n-12 per group. EO771: n = 10 per group.
(PDF)

**S2 Fig. Average daily running distance is not correlated with time to euthanasia, perfused vessel count, hypoxic fraction or CD31[+] vessel count.** Correlation of the time to euthanasia (due to maximum tumour size) with average daily running distance in mice with B16-F10 **(a)** or EO771 **(b)** tumours. Correlation of perfused vessel number with average daily running distance in mice with B16-F10 **(c)** or EO771 **(d)** tumours. Correlation of hypoxic area with average daily running distance in mice with B16-F10 **(e)** or EO771 **(f)** tumours. Correlation of CD31[+] vessel number with average daily running distance in mice with B16-F10 **(g)** or EO771 **(h)** tumours. Data analysed by Pearson (b, c, f, g, h) or Spearman correlation (a, d, e). Data shown as scatter plot with best fit line with 95% CI bands. B16-F10: n = 10–12, EO771: n = 9–10.
(PDF)

**S3 Fig. Individual weight change curves for mice bearing B16-F10 or EO771 tumours.** Individual body weight change over time for non-exercising **(a, b)** and exercising **(c, d)** mice bearing B16-F10 **(a, c)** or EO771 tumours **(b, d)**. Weight change percentage uncorrected for tumour weight. B16-F10 body weight change: n = 12 per group; EO771 body weight change: n = 11–12.
(PDF)

**S4 Fig. Weight loss after implant is associated with a shorter tumour lag phase and shorter survival in mice with EO771 breast cancer.** Tumour establishment time (lag phase, time to 175 or 100 mm$^3$) in mice with B16-F10 **(a)** or EO771 **(b)** tumours according to mouse weight change. Exponential tumour growth rate (time for the tumour to quadruple in volume) in mice that did or did not lose weight with B16-F10 **(c)** or EO771 **(d)** tumours. Data are shown as individual data points and mean ± 95% CI. Data analysed using a two-tailed students t test. B16-F10 lag phase weight loss: n = 9, no weight loss: n = 15; EO771 lag phase weight loss: n = 8, no weight loss: n = 11; B16-F10 exponential phase weight loss: n = 7, no weight loss: n = 12; EO771 exponential phase weight loss: n = 6, no weight loss: n = 11. Survival curves for mice with or without weight loss while bearing B16-F10 **(e)** or EO771 **(f)** tumours. Animals

were included in survival analysis only if euthanasia was due to tumour burden. Data analysed using Log-rank test.
(PDF)

**S5 Fig. Representative B16-F10 and EO771 tumour sections stained for CD31, pimonidazole and with Hoechst 33342.** Red: CD31, green: pimonidazole, blue: Hoechst 33342. Closed arrows indicate examples of perfused CD31$^+$ vessels and open arrows indicate examples of unperfused CD31$^+$ vessels. Images are at 20x or 40x magnification as indicated.
(PDF)

**S6 Fig. Correlations between hypoxia, perfusion and CD31$^+$ vessel density in B16-F10 and EO771 tumours.** Correlation of perfused vessel number with CD31$^+$ vessel number in B16-F10 **(a)** or EO771 **(b)** tumours. Correlation of hypoxia with CD31$^+$ vessel number in B16-F10 **(c)** or EO771 **(d)** tumours. Correlation of hypoxic area with perfused vessel number in B16-F10 **(e)** or EO771 **(f)** tumours. Data analysed by Pearson (a, b, d, f) or Spearman correlation (c, e). Data shown as scatter plot with best fit line with 95% CI bands. B16-F10: perfused vs CD31 vessels n = 16; hypoxia vs CD31 vessels n = 18; hypoxia vs perfusion n = 22; EO771: perfused vs CD31 vessels n = 25; hypoxia vs CD31 vessels n = 18; hypoxia vs perfusion n = 19.
(PDF)

**S1 Data.**
(XLSX)

## Acknowledgments

We would like to acknowledge Dr. Andreas Moeller (QIMR Berghofer, Australia) for the kind gift of the EO771 cells.

## Author Contributions

**Conceptualization:** Linda A. Buss, Margaret J. Currie, Gabi U. Dachs.

**Data curation:** Linda A. Buss, Gabi U. Dachs.

**Formal analysis:** Linda A. Buss.

**Funding acquisition:** Bridget A. Robinson, Gabi U. Dachs.

**Investigation:** Linda A. Buss.

**Methodology:** Linda A. Buss.

**Project administration:** Gabi U. Dachs.

**Supervision:** Abel D. Ang, Barry Hock, Bridget A. Robinson, Margaret J. Currie, Gabi U. Dachs.

**Writing – original draft:** Linda A. Buss.

**Writing – review & editing:** Linda A. Buss, Abel D. Ang, Barry Hock, Bridget A. Robinson, Margaret J. Currie, Gabi U. Dachs.

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
