## [Decision Letter · Decision Letter 0]

9 Dec 2019

PONE-D-19-28381

Effect of post-implant exercise on tumour growth rate, perfusion and hypoxia in mice

PLOS ONE

Dear Dr. Dachs,

Thank you for submitting your manuscript to PLOS ONE. After careful consideration, we feel that it has merit but does not fully meet PLOS ONE’s publication criteria as it currently stands. Therefore, we invite you to submit a revised version of the manuscript that addresses the points raised during the review process.

While both reviewers found this an interesting study, there were a number of issues raised which should be addressed if the authors intend to submit a revised manuscript. 

We would appreciate receiving your revised manuscript by Jan 23 2020 11:59PM. To enhance the reproducibility of your results, we recommend that if applicable you deposit your laboratory protocols in protocols.io, where a protocol can be assigned its own identifier (DOI) such that it can be cited independently in the future. For instructions see: http://journals.plos.org/plosone/s/submission-guidelines#loc-laboratory-protocols

We look forward to receiving your revised manuscript.

Kind regards,

Salvatore V Pizzo

Academic Editor

PLOS ONE

Journal Requirements:

2. Please include a caption for figure 5.

This research was funded in part by the Mackenzie Charitable Foundation, NZ. We would also like to acknowledge Dr. Andreas Moeller (QIMR Berghofer, Australia) for the kind gift of the EO771 cells and Andrew Dachs (Decision Consulting Ltd, NZ) for developing the magnetic counter for the mouse wheels. The funders had no role in study design, data collection and analysis, decision to publish, or preparation of the manuscript.

We note that you received funding from a commercial source: Decision Consulting Ltd, NZ

Reviewers' comments:

Reviewer's Responses to Questions

**Comments to the Author**

1. Is the manuscript technically sound, and do the data support the conclusions?

Reviewer #1: Yes

Reviewer #2: Yes

2. Has the statistical analysis been performed appropriately and rigorously? 

Reviewer #1: Yes

Reviewer #2: Yes

3. Have the authors made all data underlying the findings in their manuscript fully available?

Reviewer #1: Yes

Reviewer #2: Yes

4. Is the manuscript presented in an intelligible fashion and written in standard English?

Reviewer #1: Yes

Reviewer #2: Yes

5. Review Comments to the Author

Reviewer #1: This is a nice study evaluating the effect of exercise (running wheel) on tumor growth and vascular characteristics using B16F10 and E0771 models. Strengths of the work include the use of orthotopic models, appropriate sample size and statistical analysis, and the use of experimental design that minimizes animal stress. Also, this work is important because it reports a non-significant impact of exercise on tumor growth/vascular biology, which is important in the current climate of exercise oncology research where negative results are often ignored. Overall, I would like to see the study published, but do have some concerns which need to be addressed.

1. Abstract Line 20 “This has only been shown in breast and prostate cancer models…” seems a little inaccurate. Several studies have shown increases in tumor vessel perfusion and/or hypoxia. Perhaps soften the wording to "has been shown most convincingly in breast and prostate cancer models..."

2. Mice received an exercise wheel on the day of tumor transplant. It’s not clear to me what clinical situation this is meant to mimic. Typically, patients are either already exercisers who then develop cancer (modeled by pre-implantation training) or will begin exercise after a cancer diagnosis (modeled by exercise beginning after a detectable/measurable tumor is present). Exercise beginning on the day of tumor cell inoculation is not an obvious model for me. Please explain in your introduction or discussion, or if there is no explanation, please list as a study limitation in the discussion.

3. Were mice that were euthanized early included in the analysis of vasculature and hypoxia?

4. Others have shown in B16F10 tumors that exercise did not change vessel density, but did change the average length of the vessels or the number of open lumens. Did the authors evaluate these parameters in either tumor model?

5. There appears to be significant variation in the distance run between individual mice. Is it possible to analyze tumor growth or vascular characteristics in a way that takes this variation into consideration? For example, perhaps tumors behave differently in low exercising vs high exercising mice. Is there any correlation between amount of daily running and survival time, hypoxia, or vessel number?

6. I am concerned about the conclusions that were drawn regarding vascular structure, function, etc. based on images that are very difficult to see and interpret. In figure 3a, the images are extremely blurry. It would be nice to see both a low magnification and high magnification image for each condition. Also, how do the authors know that the Hoecsht 3342 is demonstrating perfused tumor vessels and not leakage outside of vessels? These images would be strengthened greatly by the addition of CD31 or VE-cadherin immunoflourescent staining so that the location of Hoecsht 3342 and of hypoxyprobe-1 relative to endothelium can truly be assessed. (3 color staining showing Hoecsht, hypoxia, and endothelium together)

7. Similarly, it’s not clear to me how the hypoxic area was determined? What software was used to determine the % hypoxic area? Is this a ratio of pimonidazole positive: total nuclei or pimonidazole: total tissue area? It seems that comparing to the total nuclei would be most appropriate because some of the tissue area may be necrotic, but if this is not possible due to software limitations, please explain the process for determining the % hypoxic area.

8. The figure legend for Figure 5 is labeled as Figure 1.

Reviewer #2: This paper describes the results of a study of post-implantation exercise in syngeneic mouse tumour graft using B16-F10 melanoma cells and EO771 breast cancer cells. No differences were observed including in measures of tumour growth, and hypoxic fraction and vascular density at sacrifice.

The work appears to have been conducted carefully, and should be published – not least in the interests of avoiding publication bias. The discussion is well balanced.

I have a few relatively minor points for clarification.

In figure 1, the y-axis is labelled as cumulative survival. This is potentially a little confusing as all animals were subjected to pre-specified euthanasia end-points (I think). In this case, it would be better to simply refer to % reaching end-point. Could the authors also clarify whether all end-points were reached due to tumour size, or where some were due to weight loss or other criteria?

In figure 2, the mean distance run does not appear to be reduced with tumour growth. Does this suggest a relative low systemic effect of the malignancy? Could the authors comment?

In figure 3, the hypoxic area is extremely variable. Could the authors describe this further? In any one tumour, were the 10 random fields similar? Or was the variability observed within tumours?

In figures 3 and 4, as indicated in the legends, there are slightly different numbers. Could the authors indicate why?

6. PLOS authors have the option to publish the peer review history of their article (what does this mean?). If published, this will include your full peer review and any attached files.

Reviewer #1: No

Reviewer #2: No

---

## [Author Response · Author response to Decision Letter 0]

14 Jan 2020

The Response to Reviewers gives point-by-point response to the reviewers.

---

## [Decision Letter · Decision Letter 1]

28 Jan 2020

PONE-D-19-28381R1

Effect of post-implant exercise on tumour growth rate, perfusion and hypoxia in mice

PLOS ONE

Dear Dr. Dachs,

Thank you for submitting your manuscript to PLOS ONE. After careful consideration, we feel that it has merit but does not fully meet PLOS ONE’s publication criteria as it currently stands. Therefore, we invite you to submit a revised version of the manuscript that addresses the points raised during the review process.

As you will see, both reviewers were in agreement that this manuscript is ready to be accepted for publication.  However, Reviewer No. 2 would like to see some clarifications made in the Figure Legends. If the authors agree to this issue, then we would certainly welcome a revised manuscript ready for publication. 

We would appreciate receiving your revised manuscript by Mar 13 2020 11:59PM. To enhance the reproducibility of your results, we recommend that if applicable you deposit your laboratory protocols in protocols.io, where a protocol can be assigned its own identifier (DOI) such that it can be cited independently in the future. For instructions see: http://journals.plos.org/plosone/s/submission-guidelines#loc-laboratory-protocols

We look forward to receiving your revised manuscript.

Kind regards,

Salvatore V Pizzo

Academic Editor

PLOS ONE

Reviewers' comments:

Reviewer's Responses to Questions

**Comments to the Author**

1. If the authors have adequately addressed your comments raised in a previous round of review and you feel that this manuscript is now acceptable for publication, you may indicate that here to bypass the “Comments to the Author” section, enter your conflict of interest statement in the “Confidential to Editor” section, and submit your "Accept" recommendation.

Reviewer #1: All comments have been addressed

Reviewer #2: All comments have been addressed

2. Is the manuscript technically sound, and do the data support the conclusions?

Reviewer #1: Yes

Reviewer #2: Yes

3. Has the statistical analysis been performed appropriately and rigorously? 

Reviewer #1: Yes

Reviewer #2: Yes

4. Have the authors made all data underlying the findings in their manuscript fully available?

Reviewer #1: Yes

Reviewer #2: Yes

5. Is the manuscript presented in an intelligible fashion and written in standard English?

Reviewer #1: Yes

Reviewer #2: Yes

6. Review Comments to the Author

Reviewer #1: Congratulations on this work. I'm very excited to see it out in publication! Nice use of good models and good statistics!

Reviewer #2: (No Response)

7. PLOS authors have the option to publish the peer review history of their article (what does this mean?). If published, this will include your full peer review and any attached files.

Reviewer #1: No

Reviewer #2: No

---

## [Author Response · Author response to Decision Letter 1]

30 Jan 2020

Below are our detailed responses to the reviewer’s comment (indented in blue).

Reviewer 2.

'However they do not appear to have adjusted the manuscript to make these issues clear in the revision. For instance, in the clarifications, I suggested, for the figure legends, it would seem reasonable to make the answers to the questions explicit with a simple sentence. I do not regard this as mandatory, but do feel the authors would improve the clarity of the manuscript if they did this. Otherwise the reader will puzzle over the issues which I raised in just the same way as myself.'

We have made the following changes in response to the reviewer’s comments:

Line 200 Fig 1 legend, added: ‘mice euthanised due to other endpoints were excluded from the survival analysis’

Line 211 added: ‘and that the systemic effects are comparatively low’

Line 264 added: ‘Hypoxia was more variable between tumours than within individual tumours (mean SD of hypoxic fraction of individual fields used for analysis: 4.96 and 7.8 for B16-F10 and EO771, respectively, vs SD of mean hypoxic fraction between tumours: 7.27 and 9.69 for B16-F10 and EO771, respectively).’

Line 278 Fig 3 legend, added: ‘B16-F10 No Ex and Ex: n=12, EO771 No Ex: n=9 (two mice had intra-peritoneal tumours and one tumour exhibited low-level, diffuse perfusion which could not be quantified) and Ex: n=10 (one mouse had intra-peritoneal tumours, and one was euthanised before tumour development due to pyometra).’

Line 292 Fig 4 legend, added: ’ B16-F10 No Ex: n=8 (four tumours could not be quantified due to dark pigmentation), B16-F10 Ex: n=10 (two tumours could not be quantified due to dark pigmentation), EO771 No Ex: n=10 (two mice had intra-peritoneal tumours) and Ex: n=10 (one mouse had intra-peritoneal tumours, and one pyometra).’

Line 311 Fig 5 legend, added: ‘EO771: n=10 per group (three mice had intra-peritoneal tumours, and one pyometra).’

Line 331 Fig 6 legend, added: ‘B16-F10: n=24, EO771: n=20 (three mice had intra-peritoneal tumours, and one pyometra). (b) Comparison of the number of CD31+ vessels in B16-F10 and EO771 tumours. B16-F10: n=18 (six tumours could not be quantified due to dark pigmentation), EO771: n=20 (three mice had intra-peritoneal tumours, and one pyometra). Quantification of hypoxia (c) and perfusion (d) in B16-F10 vs EO771 tumours. B16-F10: n=24, EO771: n=20 (three mice had intra-peritoneal tumours, and one pyometra).’

---

## [Editor Report · Decision Letter 2]

4 Feb 2020

Effect of post-implant exercise on tumour growth rate, perfusion and hypoxia in mice

PONE-D-19-28381R2

Dear Dr. Dachs,

We are pleased to inform you that your manuscript has been judged scientifically suitable for publication and will be formally accepted for publication once it complies with all outstanding technical requirements.

With kind regards,

Salvatore V Pizzo

Academic Editor

PLOS ONE
---

## [Editor Report · Acceptance letter]

20 Feb 2020

PONE-D-19-28381R2 

Effect of post-implant exercise on tumour growth rate, perfusion and hypoxia in mice 

Dear Dr. Dachs:

I am pleased to inform you that your manuscript has been deemed suitable for publication in PLOS ONE. Congratulations! Your manuscript is now with our production department. 

With kind regards,

on behalf of

Dr. Salvatore V Pizzo 

Academic Editor

PLOS ONE